# State and Strategy of Production, Market and Integrated Management of Mineral Water, South Korea

**Byeong Dae Lee [1], Yong Hwa Oh [1], Won Bin Kim [1], Jaehong Hwang [1], Woo-Ri Lim [2], Sung-Ja Choi [1] and Se-Yeong Hamm [2],***

[1]  Groundwater Research Center, Korea Institute of Geoscience and Mineral Resources, Daejeon 34132, Korea; blee@kigam.re.kr (B.D.L.); yonghwa.oh@gmail.com (Y.H.O.); onebean@kigam.re.kr (W.B.K.); hwangjh@kigam.re.kr (J.H.); sjchoi@kigam.re.kr (S.-J.C.)
[2]  Department of Geological Sciences, Pusan National University, Busan 46241, Korea; wooriful@naver.com
*  Correspondence: hsy@pusan.ac.kr; Tel.: +82-51-510-2252

**Abstract:** In this study, the state of the production, market, and integrated management of mineral water in South Korea was assessed and a strategy was established to improve the competitiveness of mineral water with respect to foreign brands. Diverse data, including a number of mineral water manufacturers, daily permitted volume, daily production amounts, sales volume, and sales amounts, were used to clarify the state of mineral water and establish a strategy. Real-time data transmitted from the mineral water manufacturers were analyzed and managed in an integrated management system (IMS). Chemical characteristics of mineral water were analyzed to assess water quality and identify problems related to contamination. This study shows that specialization plans according to geological characteristics should be established, rather than focusing on quantitative growth by a low-cost sales method. The establishment of contamination management plans was found to be needed for wells with potential pollution due to sub-surface inflow into the wells. We suggest that an independent editing system of the attribute information needs to be designed based on the analysis and management of the IMS.

**Keywords:** mineral water; permitted volume; production amounts; sales volume; integrated management system

## 1. Introduction

Water is essential for life and is a valuable natural resource. The acquisition of fresh water has been a key factor influencing human settlements throughout the history of mankind. Mineral water is commonly produced from bedrock aquifers in which inorganic constituents have been dissolved in groundwater through a water–mineral interaction. This process eliminates ingredients and impurities that are harmful to humans during the movement of groundwater through the bedrock. As a consequence, mineral water contains various mineral components and specific trace elements depending on the geologic property of the aquifers [1], and is hygienically safe with little change in water quality.

At the present time, mineral water is commonly used by consumers and can be easily purchased at supermarkets and large discount stores. Minerals are required by human beings for nutrition, growth, sustaining body functions, and well-being. In Korea, mineral water originates from groundwater with sufficient quality to meet the standard for drinking water [2]. For this reason, the mineral water must be secured in a clean and safe state to preserve its quality. It is also necessary to clarify the state and various problems of the water and establish a management strategy that includes geological

characteristics, production amounts, production and monitoring wells, the source water environment, and an automatic measurement system of monitoring wells.

Since 1995, the Korean government, via the Ministry of Environment, has managed mineral water under the Drinking Water Management Act (DWMA) [3]. According to the DWMA, only mineral water is allowed as the water source for manufacturing bottled waters [4]. In 1997, the first nationwide investigation into mineral waters was under taken by the South Korea Institute of Geoscience and Mineral Resources (KIGAM) for ensuring a high quality and better understanding of the hydrochemical properties of mineral water [5]. A total of 87 mineral water manufacturers were officially registered in 2006, but 62 companies were in action as of December 2018. Article 3 of Korea's DWMA classifies drinking water into drinkable natural mineral water, drinkable treated tap water, drinkable spring water, drinkable salt groundwater, and drinkable deep ocean water. Natural mineral water is the groundwater or spring water originating from bedrock aquifers, which ensure high and safe water quality. According to the DWMA, mineral water is the product of drinking water manufactured exclusively by physical treatment. The use of chemical treatments for original groundwater used for bottled waters are prohibited [6]. KIGAM, the groundwater specialized agency designated by Korea's Ministry of Environment (MOE), has monitored the quantity and quality of mineral water based on real-time data transmitted by mineral water manufacturers since 1999, analyzed anomalous data, and helped to establish countermeasures. In addition, KIGAM has operated an integrated management system (IMS) for the management of mineral water through a database of information on the manufacturers, pumping and monitoring wells, water quantity and quality, rock/soil, and geological data. On the other hand, apart from data transmission based on the real-time, the mineral water manufacturers should hand in measurement data for the quantity and quality of water to the local government monthly according to Article 22 of the DWMA.

In South Korea, the National Groundwater Monitoring Network (NGMN) has operated since 1995, earlier than the on-line system for mineral water [7]. The goal of the NGMN is to establish 320 groundwater monitoring stations, and the main purpose of these stations is to monitor long-term general trends in water level fluctuations and groundwater quality throughout South Korea. The National Institute of Environmental Research (NIER), under MOE, conducted the Safe Groundwater Supply Project (SGSP) from 2014 to 2016, and proposed a roadmap for groundwater quality management and a supply plan for safe drinking groundwater [8].

The purpose of this study is to present the current state of the mineral water production, market, quality, quantity, and integrated management in South Korea, and then to establish a strategy to improve the competitiveness with respect to foreign brands.

## 2. Materials and Methods

### 2.1. Geological Settings of Mineral Water Manufacturers

Geology is an important factor in determining the quality of natural mineral water. The geology of South Korea is complicated because of complex tectonic events and a wide span of ages, ranging from the Archean to Quaternary Holocene epochs (Figure 1). The geology of the areas where mineral water factories are located is mostly composed of Precambrian metamorphic rocks, Mesozoic granite, Ogcheon metamorphic rocks, and Quaternary Jeju volcanic rocks.

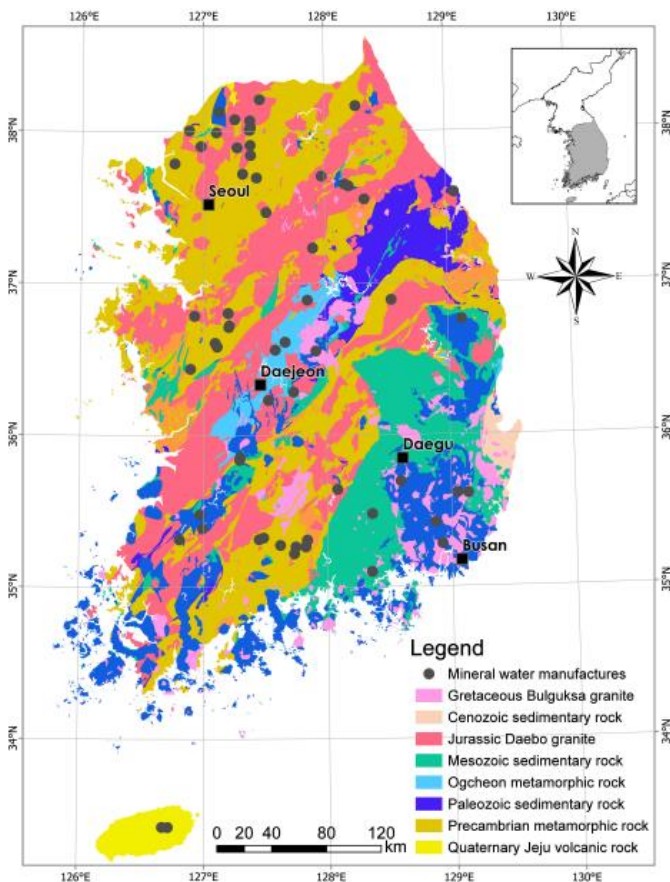

**Figure 1.** Geological map of South Korea and locations of mineral water bottlers facilities.

Of the 62 mineral water factories operating in South Korea, 26 are distributed in areas characterized by Precambrian metamorphic rocks. Precambrian metamorphic rocks, one of the most widespread geological units in South Korea, are generally composed of granitic gneiss, banded gneiss, porphyroblastic gneiss, orbicular gneiss, and banded biotite gneiss [9]. Each of the factories in the Precambrian metamorphic rocks area is located on the granitic gneiss. Granitic gneiss is predominant in the Precambrian metamorphic rocks according to its petrochemical properties and spatial distribution.

Twenty-three mineral water factories are located in areas of Mesozoic granite. Mesozoic granite shows an equigranular medium to coarse grained texture with the major constituent minerals of quartz, plagioclase, orthoclase, microcline, and biotite. Mesozoic granite is classified into two types of granite, namely, Jurassic Daebo granite and Cretaceous Bulguksa granite. Daebo granite is considered to have been formed by partial melting of the lower continental crust, and solidified at relatively deep levels [10,11]. Bulguksa granite is considered to have been formed by partial melting of the mantle and solidified at relatively shallow depths [10–12].

Eleven mineral water factories are located in the Ogcheon metamorphic belt distributed mostly in the central part of South Korea. Rocks in the Ogcheon metamorphic belt are composed of various metamorphic sedimentary rocks, such as chlorite schist, sandy phyllite, biotite schist, quartzite, and quartz-sericite schist [13,14]. The main constituent minerals are quartz, dolomite, biotite, muscovite, chlorite, calcite, silicate, and various metamorphic minerals.

Two mineral water factories are located on the eastern part of Jeju island. Jeju, the biggest island in South Korea, is the site of active quaternary volcanoes [15]. Jeju Island consists of various volcanic rocks, including predominantly alkali basalt in lowland areas, and lesser trachybasalt, trachyandesite, and trachyte in the highlands. In addition, at least 400 scoria cones, phreatomagmatic tuff rings, and tuff cones are widespread on the island, intercalated with shallow marine deposits [16]. The main

constituent minerals of volcanic rocks are olivine, pyroxene, hornblende, biotite, feldspars, and volcanic glass. The water resources on the island rely entirely on groundwater because there is no surface water, such as from streams [17]. Groundwater levels in the northern and southern basins are generally higher than those in the eastern and western basins on the island [18].

*2.2. Database Preparation*

All data, including a number of mineral water manufacturers in each province, daily permitted volume, daily production amounts, sales volume, and sales amounts, were obtained from the Korea Mineral Water Association (KMWA) to identify the state of mineral water in South Korea and establish a strategy to improve the brand value of products through data analysis. We analyzed the ratio of production amounts to the permitted volume, the relationship between sales volume and sales amounts, the sales volume increase rate, the sales amount increase rate, and the ratio of manufacturer numbers in each geology and province using these data. Data for the IMS were collected from information transmitted in real-time from the mineral water factories. The information transmitted was analyzed and managed in the IMS, which consists of metadata and a dataset of various attributes.

Mineral water samples were collected directly from the production wells of the 62 mineral water factories from March to June 2017. The samples were taken after temperature, pH, and electrical conductivity (EC) had been stabilized, then filtered through a 0.45 $\mu$m membrane filter. Temperature, pH, EC, and alkalinity as bicarbonate ($HCO_3$) were measured in situ. Samples for cation analysis were acidified to pH < 2 with ultrapure $HNO_3$. Each water sample was analyzed to determine the concentration (as mg/L) of major and minor ions (K, Na, Ca, Mg, $SiO_2$, F, Cl, $SO_4$, and $NO_3$) using inductively coupled plasma atomic emission spectrometry (ICP-AES, Optima 4300 DV, PerkinElmer, Waltham, MA, USA) and ion chromatography (IC, DX-1500, Dionex, Sunnyvale, CA, USA) at KIGAM laboratory, respectively.

*2.3. Statistical Methods*

Considering that the aim of this study is to present the current state of the mineral water in South Korea and to establish a strategy, descriptive statistical methods are better than inductive statistical methods. So, we used simple descriptive statistical methods, such as maximum, minimum, standard deviation, and representative value including arithmetic mean and median, to present data on the production, market, and hydrochemical characteristics of mineral water.

*2.4. Integrated Management*

An integrated management system (IMS) of mineral water based on a geographical information system (GIS) was designed and built to store and efficiently manage data transmitted from the mineral water manufacturers (Figure 2). In order to store and manage real-time data collected from all manufacturers, it is not efficient to put all information directly into one database to be managed. Therefore, all information used in this system is kept as a file, and the metadata is used to efficiently manage the data constructed in the system. This metadata is designed to efficiently manage various types of data by making it accessible through queries, such as basic structured query language (SQL) in the form of database tables.

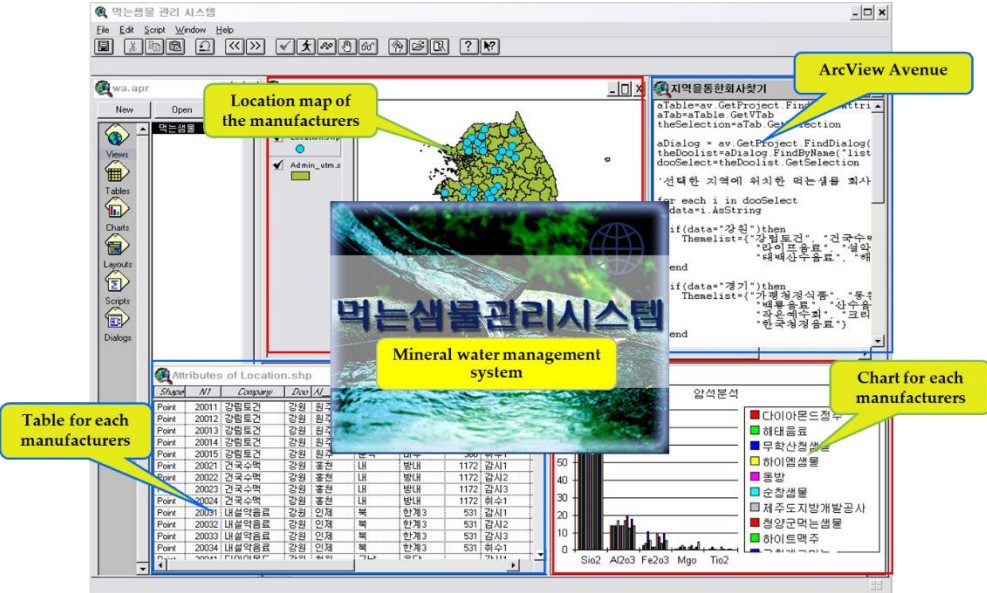

**Figure 2.** The initial screen of the mineral water management system.

2.4.1. Acquisition and Transmission of Mineral Water Data

Most mineral water manufacturers in Korea have three to five production wells each, with at least three monitoring wells. All manufacturers have to install and operate an automated monitoring system in their monitoring wells, and a flowmeter in the production wells. The monitoring system consists of a well, a probe, a transducer cable, a data logger unit, and a power unit. The probe in the monitoring well continuously measures the water level, hydrogen ion concentration (pH), electrical conductivity (EC), and temperature (T), and the flowmeter records the extraction rate. The data logger is a device that stores measured data and transmits the stored data to KIGAM's network server using on-line and wireless communication, on an hourly basis (Figure 3). The data stored in the data logger are transmitted using the transmission control protocol/internet protocol (TCP/IP) remote communication method.

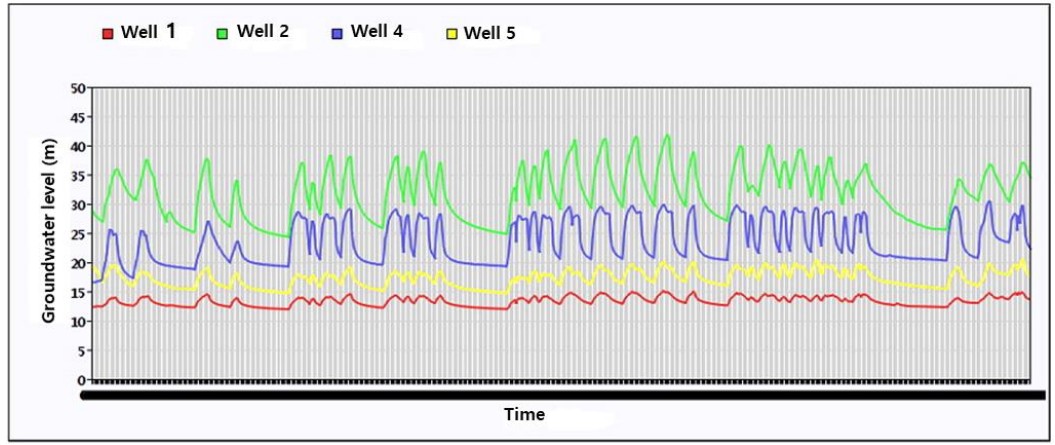

**Figure 3.** Mineral water level data (monitoring well 1 of red color, monitoring well 2 of green color, monitoring well 4 of violet color, and monitoring well 5 of yellow color) are transmitted from the mineral water manufacturers based on real-time. Horizontal axis is the measurement time and vertical axis is the water level value. Numbers represent depth to water (unit: m).

2.4.2. IMS Design

The basic structure of the IMS for the mineral water consists of applications for user access, metadata for the managing and accessing of collected data, and a dataset of various attributes. Database supporting files such as application metadata are accessed and managed through structured query language (SQL) using ActiveX Data Objects (ADO), and other images are displayed through the access path stored in the metadata. The information of the metadata used in the IMS includes information additional to the basic data and an access path for accessing the data in the IMS. The metadata consists of database tables in the form of Microsoft Access (mdb) files and the database table stores information for accessing and explaining the data for each menu.

The collected and managed data largely consist of basic information regarding the manufacturers, the status of monitoring and pumping wells, rock/soil information, groundwater level data of the monitoring wells, the management status of the pumping wells, the extraction rate, and water quality data. The manufacturers' basic information comprises the manufacturer name, telephone number, address, geology, and original equipment manufacturing (OEM), as well as photos related to the manufacturers. Photos of the manufacturers can be displayed through a search function. The data of the monitoring and pumping wells held by the manufacturers are stored as attribute data, so that users can browse the relevant information through inquiries. Rock and soil information includes the physical and chemical analysis data on rocks and soils of the manufacturers, and additional information on the analytical components can be manually entered. Water level data on the monitoring wells include actual and automatic measurement data by measurement date. We actually measure the water level by hand monthly in order to check for errors between the actual and automatic measurements. If an error is found in the groundwater level, the management system calculates and provides information to the user. The water quality data of the manufacturers are stored and can be updated.

*2.5. Uncertainties and Shortcomings*

The data such as daily permitted volume, daily production amounts, sales volume, and sales amounts used in this study were handed in to KMWA by each mineral water manufacturer, so there are some uncertainties. Although the current IMS has the flexibility to modify, add, and update data, it has a shortcoming in that a large amount of information should be newly inputted in accordance with the designed form because it has not been rebuilt since March 2010.

**3. Results and Discussion**

*3.1. Chemical Characteristics of Mineral Water*

Understanding groundwater quality is important as it is the main factor determining its suitability for drinking and domestic, agricultural, and industrial purposes [19]. Tables 1 and 2 represent the physicochemical properties of mineral water and statistical summary of the analytical data for each geology, respectively.

**Table 1.** Physicochemical compositions of groundwater for mineral waters for different types of rocks (PMR: Precambrian metamorphic rocks, MG: Mesozoic granite, OMR: Ogcheon metamorphic rocks, OJV: Quaternary Jeju volcanic rocks). Concentration unit is mg/L unless otherwise noted. Electrical conductivity (EC) in μS/cm.

| Samples | K | Na | Ca | Mg | SiO$_2$ | F | Cl | SO$_4$ | NO$_3$-N | HCO$_3$ | EC | pH | T (°C) |
|---|---|---|---|---|---|---|---|---|---|---|---|---|---|
| PMR1 | 0.63 | 2.07 | 5.49 | 1.02 | 6.97 | 0.17 | 1.81 | 4.90 | 1.67 | 19.32 | 57.40 | 7.34 | 8.03 |
| PMR2 | 2.26 | 3.31 | 9.63 | 1.62 | 8.80 | 0.12 | 4.50 | 10.04 | 4.06 | 17.08 | 108.42 | 7.43 | 9.92 |
| PMR3 | 1.12 | 8.98 | 20.06 | 5.39 | 21.70 | 0.60 | 8.52 | 7.06 | 0.90 | 95.74 | 166.96 | 7.76 | 17.38 |
| PMR4 | 0.64 | 2.92 | 8.24 | 1.91 | 15.47 | 0.08 | 1.92 | 2.19 | 0.79 | 39.14 | 76.83 | 7.76 | 14.10 |
| PMR5 | 0.57 | 5.18 | 21.53 | 1.23 | 13.50 | 0.59 | 3.38 | 5.03 | 0.86 | 79.30 | 142.50 | 7.77 | 13.75 |
| PMR6 | 1.63 | 14.10 | 48.84 | 7.82 | 26.47 | 0.96 | 3.75 | 14.43 | 0.18 | 197.25 | 349.53 | 7.00 | 14.41 |
| PMR7 | 1.20 | 17.80 | 37.43 | 3.46 | 16.13 | 0.69 | 14.57 | 30.67 | 0.33 | 110.56 | 303.50 | 6.99 | 15.88 |
| PMR8 | 0.85 | 9.47 | 24.34 | 0.73 | 12.20 | 1.12 | 3.79 | 13.11 | 0.89 | 91.01 | 179.46 | 7.74 | 12.95 |
| PMR9 | 0.76 | 5.12 | 5.36 | 1.15 | 24.78 | 0.12 | 1.55 | 1.63 | 0.46 | 35.84 | 70.20 | 6.95 | 13.70 |
| PMR10 | 0.97 | 4.25 | 6.78 | 1.06 | 15.84 | 0.15 | 2.48 | 5.48 | 2.57 | 26.84 | 73.22 | 6.59 | 12.05 |
| PMR11 | 1.67 | 8.55 | 23.46 | 5.79 | 27.44 | 0.28 | 4.03 | 3.16 | 0.47 | 121.41 | 202.02 | 6.65 | 15.30 |
| PMR12 | 0.83 | 3.69 | 8.70 | 2.99 | 9.77 | 0.05 | 2.38 | 8.38 | 0.63 | 41.18 | 86.53 | 7.39 | 9.85 |
| PMR13 | 1.91 | 10.73 | 24.00 | 5.72 | 29.23 | 0.19 | 5.83 | 7.19 | 0.91 | 118.95 | 232.70 | 6.87 | 15.00 |
| PMR14 | 1.39 | 13.08 | 25.04 | 5.00 | 32.50 | 0.25 | 8.34 | 8.90 | 1.98 | 98.88 | 228.26 | 6.93 | 14.92 |
| PMR15 | 1.08 | 6.88 | 24.27 | 4.35 | 15.40 | 0.12 | 2.99 | 11.75 | 0.01 | 99.63 | 173.40 | 7.54 | 17.80 |
| PMR16 | 0.90 | 6.13 | 14.23 | 1.65 | 14.90 | 0.12 | 3.45 | 13.18 | 0.19 | 54.30 | 128.38 | 7.21 | 14.68 |
| PMR17 | 1.68 | 5.08 | 14.16 | 2.17 | 13.34 | 0.49 | 1.83 | 8.45 | 1.05 | 62.70 | 120.94 | 7.92 | 14.28 |
| PMR18 | 2.06 | 6.03 | 19.00 | 1.70 | 19.90 | 0.15 | 3.4 | 11.0 | 0.25 | 76 | 157 | 7.57 | 16 |
| PMR19 | 0.84 | 4.25 | 9.92 | 1.04 | 17.67 | 0.19 | 2.60 | 3.75 | 1.44 | 46.26 | 102.30 | 8.13 | 17.33 |
| PMR20 | 0.79 | 3.84 | 7.33 | 0.90 | 20.64 | 0.13 | 1.56 | 2.48 | 0.68 | 28.68 | 59.22 | 7.24 | 15.86 |
| PMR21 | 2.03 | 10.80 | 21.76 | 5.10 | 32.36 | 0.35 | 5.68 | 6.92 | 0.96 | 108.34 | 207.02 | 6.79 | 15.58 |
| PMR22 | 0.49 | 4.83 | 4.33 | 0.41 | 8.73 | 0.27 | 1.51 | 6.08 | 0.16 | 23.64 | 55.76 | 7.33 | 11.93 |
| PMR23 | 1.13 | 5.46 | 13.95 | 12.23 | 7.19 | 0.19 | 3.31 | 9.46 | 1.02 | 101.31 | 202.24 | 7.70 | 14.32 |
| PMR24 | 0.75 | 8.87 | 26.46 | 4.17 | 17.84 | 0.12 | 8.19 | 12.34 | 0.42 | 100.35 | 206.38 | 7.27 | 14.32 |
| PMR25 | 2.75 | 9.93 | 27.00 | 5.14 | 19.88 | 0.67 | 4.62 | 8.38 | 0.51 | 122.00 | 234.33 | 7.33 | 16.00 |
| PMR26 | 2.67 | 16.16 | 34.64 | 3.71 | 21.60 | 1.03 | 9.21 | 10.71 | 0.01 | 148.23 | 271.00 | 7.61 | 17.18 |
| MG1 | 1.06 | 2.86 | 13.50 | 3.40 | 9.19 | 0.73 | 1.4 | 6.8 | 1.58 | 55 | 154 | 7.07 | - |
| MG2 | 0.44 | 9.98 | 7.67 | 1.20 | 41.76 | 0.12 | 3.19 | 4.21 | 0.08 | 41.42 | 87.42 | 7.10 | 17.93 |
| MG3 | 0.51 | 17.39 | 17.84 | 5.65 | 11.24 | 0.65 | 3.66 | 14.81 | 1.98 | 113.22 | 221.42 | 8.19 | 14.63 |
| MG4 | 0.81 | 9.46 | 25.73 | 1.79 | 23.74 | 0.30 | 3.83 | 8.14 | 0.49 | 106.15 | 208.66 | 7.19 | 15.36 |
| MG5 | 0.82 | 6.76 | 10.07 | 1.52 | 25.56 | 0.42 | 4.31 | 2.91 | 0.67 | 49.29 | 103.26 | 7.96 | 16.15 |
| MG6 | 0.58 | 16.74 | 19.34 | 1.57 | 22.10 | 1.65 | 4.58 | 4.64 | 0.93 | 92.72 | 191.36 | 7.10 | 13.70 |
| MG7 | 0.28 | 12.54 | 20.66 | 0.67 | 20.20 | 1.14 | 4.06 | 26.44 | 0.12 | 65.88 | 174.18 | 7.75 | 18.92 |
| MG8 | 1.06 | 8.07 | 17.34 | 2.24 | 23.26 | 0.26 | 3.80 | 12.67 | 1.94 | 62.77 | 148.20 | 6.81 | 14.65 |
| MG9 | 0.40 | 6.95 | 9.94 | 1.86 | 30.46 | 0.29 | 3.04 | 3.20 | 0.22 | 57.65 | 106.58 | 7.28 | 16.04 |
| MG10 | 0.46 | 18.70 | 34.43 | 1.51 | 22.85 | 1.18 | 23.22 | 6.15 | 0.71 | 127.34 | 198.63 | 7.56 | 16.03 |
| MG11 | 0.63 | 8.08 | 14.66 | 2.32 | 8.65 | 0.24 | 1.95 | 8.01 | 1.34 | 73.26 | 152.08 | 7.43 | 13.82 |
| MG12 | 1.67 | 4.95 | 13.10 | 4.08 | 21.10 | 0.38 | 2.5 | 5.5 | 0.62 | 59 | 120 | 7.92 | 16.7 |
| MG13 | 0.52 | 5.64 | 11.20 | 1.53 | 18.80 | 0.23 | 2.12 | 1.33 | 1.00 | 51.48 | 93.26 | 7.17 | 14.63 |
| MG14 | 0.85 | 9.52 | 17.74 | 3.43 | 30.98 | 0.34 | 4.47 | 12.17 | 1.90 | 75.95 | 161.70 | 6.57 | 14.24 |
| MG15 | 0.59 | 12.46 | 11.68 | 1.44 | 29.72 | 0.29 | 7.03 | 6.76 | 4.33 | 50.02 | 138.10 | 6.43 | 15.44 |
| MG16 | 0.41 | 17.70 | 18.30 | 0.33 | 22.13 | 1.43 | 37.30 | 2.87 | 0.10 | 34.53 | 194.48 | 7.12 | 16.23 |
| MG17 | 0.37 | 6.83 | 12.50 | 1.33 | 19.46 | 0.70 | 1.34 | 3.08 | 0.32 | 64.36 | 102.76 | 7.54 | 12.50 |
| MG18 | 1.21 | 4.63 | 11.46 | 0.89 | 16.46 | 0.33 | 1.33 | 3.12 | 0.59 | 48.19 | 86.80 | 7.74 | 14.96 |
| MG19 | 0.80 | 6.02 | 13.66 | 1.44 | 15.54 | 0.17 | 4.89 | 7.62 | 2.33 | 46.67 | 117.80 | 7.05 | 12.15 |
| MG20 | 0.11 | 5.23 | 22.50 | 0.83 | 26.70 | 0.07 | 2.3 | 40.4 | 0.04 | 38 | 142 | 7.89 | 16.8 |
| MG21 | 0.10 | 4.74 | 16.72 | 0.84 | 22.98 | 0.07 | 2.15 | 26.68 | 0.06 | 39.04 | 134.10 | 7.81 | 16.68 |
| MG22 | 0.43 | 5.25 | 12.44 | 0.95 | 18.38 | 0.85 | 2.99 | 5.88 | 1.16 | 41.48 | 100.90 | 7.14 | 13.93 |
| MG23 | 0.53 | 4.74 | 8.67 | 1.43 | 21.00 | 0.13 | 1.80 | 1.00 | 1.04 | 45.87 | 88.52 | 7.42 | 11.04 |
| OMR1 | 1.21 | 13.60 | 31.76 | 13.52 | 13.90 | 0.49 | 4.22 | 24.64 | 0.06 | 167.75 | 315.60 | 7.82 | 15.42 |
| OMR2 | 1.88 | 6.04 | 50.58 | 7.69 | 31.28 | 0.73 | 2.93 | 80.32 | 0.55 | 106.37 | 353.50 | 7.63 | 15.58 |
| OMR3 | 1.36 | 3.78 | 34.30 | 3.63 | 13.00 | 0.05 | 6.83 | 3.14 | 2.24 | 113.48 | 199.50 | 7.51 | 13.80 |
| OMR4 | 1.26 | 4.83 | 33.70 | 18.20 | 11.50 | 0.17 | 3.1 | 33.1 | 0.13 | 171 | 323 | 7.78 | 14.2 |
| OMR5 | 1.27 | 7.79 | 39.04 | 8.97 | 19.98 | 0.14 | 10.12 | 32.05 | 0.71 | 130.54 | 266.18 | 6.73 | 16.44 |
| OMR6 | 0.65 | 7.77 | 41.43 | 14.33 | 11.00 | 0.13 | 19.23 | 19.97 | 1.34 | 150.98 | 329.00 | 7.28 | 12.80 |
| OMR7 | 1.46 | 3.63 | 21.82 | 2.47 | 10.87 | 0.17 | 5.60 | 7.29 | 1.68 | 74.42 | 119.18 | 7.61 | 13.06 |
| OMR8 | 0.85 | 4.46 | 30.73 | 3.20 | 9.97 | 0.09 | 2.42 | 25.64 | 0.00 | 92.01 | 210.33 | 8.10 | 13.20 |
| OMR9 | 0.83 | 8.07 | 31.05 | 12.05 | 12.35 | 0.33 | 3.77 | 23.79 | 0.45 | 137.25 | 261.00 | 7.64 | 15.15 |
| OMR10 | 1.09 | 7.24 | 24.09 | 5.48 | 24.50 | 0.32 | 7.90 | 13.44 | 0.87 | 110.06 | 224.74 | 7.65 | 14.44 |
| OMR11 | 1.33 | 7.26 | 42.20 | 11.96 | 11.44 | 0.18 | 8.44 | 18.13 | 1.52 | 149.45 | 303.33 | 7.40 | 15.90 |
| QJV1 | 2.09 | 5.30 | 2.99 | 2.28 | 26.34 | 0.06 | 5.78 | 1.59 | 0.11 | 36.91 | 80.34 | 7.74 | 14.72 |
| QJV2 | 2.56 | 7.49 | 6.40 | 6.79 | 36.50 | 0.06 | 8.52 | 1.87 | 0.62 | 63.20 | 141.14 | 7.35 | 16.04 |

**Table 2.** Statistical comparison of physicochemical components in groundwater used for bottled waters, with respect to rocks (Precambrian metamorphic rocks (PMR), Mesozoic granite (MG), Ogcheon metamorphic rocks (OMR), and Quaternary Jeju volcanic rocks (OJV)). Min: minimum, Max: maximum, Mean: average, Med: median, Std: standard deviation. Units are the same as in Table 1.

| | | K | Na | Ca | Mg | $SiO_2$ | F | Cl | $SO_4$ | $NO_3$-N | $HCO_3$ | EC | pH | T (°C) |
|---|---|---|---|---|---|---|---|---|---|---|---|---|---|---|
| PMR | Max. | 2.75 | 17.80 | 48.84 | 12.23 | 32.50 | 1.12 | 14.57 | 30.67 | 4.06 | 197.25 | 349.53 | 8.13 | 17.80 |
| | Min. | 0.49 | 2.07 | 4.33 | 0.41 | 6.97 | 0.05 | 1.51 | 1.63 | 0.01 | 17.08 | 55.76 | 6.59 | 8.03 |
| | Mean | 1.29 | 7.60 | 18.69 | 3.36 | 18.09 | 0.35 | 4.43 | 8.72 | 0.90 | 79.39 | 161.37 | 7.34 | 14.33 |
| | Med. | 1.10 | 6.08 | 19.53 | 2.58 | 16.90 | 0.19 | 3.44 | 8.38 | 0.73 | 85.16 | 161.98 | 7.34 | 14.54 |
| | Std. | 0.65 | 4.20 | 11.09 | 2.72 | 7.41 | 0.31 | 3.09 | 5.76 | 0.89 | 45.00 | 80.27 | 0.41 | 2.41 |
| MG | Max. | 1.67 | 18.70 | 34.43 | 5.65 | 41.76 | 1.65 | 37.30 | 40.40 | 4.33 | 127.34 | 221.42 | 8.19 | 18.92 |
| | Min. | 0.10 | 2.86 | 7.67 | 0.33 | 8.65 | 0.07 | 1.33 | 1.00 | 0.04 | 34.53 | 86.80 | 6.43 | 11.04 |
| | Mean | 0.64 | 8.92 | 15.70 | 1.84 | 21.84 | 0.52 | 5.53 | 9.32 | 1.02 | 62.60 | 140.31 | 7.36 | 15.11 |
| | Med. | 0.58 | 6.89 | 14.16 | 1.53 | 21.60 | 0.32 | 3.31 | 6.45 | 0.81 | 56.27 | 140.20 | 7.31 | 14.96 |
| | Std. | 0.36 | 4.76 | 6.12 | 1.24 | 7.41 | 0.45 | 8.20 | 9.64 | 1.00 | 25.22 | 42.20 | 0.45 | 3.64 |
| OMR | Max. | 1.88 | 13.60 | 50.58 | 18.20 | 31.28 | 0.73 | 19.23 | 80.32 | 2.24 | 170.80 | 353.50 | 8.10 | 16.44 |
| | Min. | 0.65 | 3.63 | 21.82 | 2.47 | 9.97 | 0.05 | 2.42 | 3.14 | 0.00 | 74.42 | 119.18 | 6.73 | 12.80 |
| | Mean | 1.20 | 6.77 | 34.61 | 9.23 | 15.43 | 0.25 | 6.77 | 25.59 | 0.87 | 127.55 | 264.12 | 7.56 | 14.54 |
| | Med. | 1.26 | 7.24 | 33.70 | 8.97 | 12.35 | 0.17 | 5.60 | 23.79 | 0.71 | 130.54 | 266.18 | 7.63 | 14.44 |
| | Std. | 0.34 | 2.81 | 8.30 | 5.20 | 6.88 | 0.20 | 4.85 | 20.43 | 0.74 | 31.08 | 70.35 | 0.35 | 1.24 |
| QJV | Max. | 2.56 | 7.49 | 6.40 | 6.79 | 36.50 | 0.06 | 8.52 | 1.87 | 0.62 | 63.20 | 141.14 | 7.74 | 16.04 |
| | Min. | 2.09 | 5.30 | 2.99 | 2.28 | 26.34 | 0.06 | 5.78 | 1.59 | 0.11 | 36.91 | 80.34 | 7.35 | 14.72 |
| | Mean | 2.32 | 6.39 | 4.70 | 4.54 | 31.42 | 0.06 | 7.15 | 1.73 | 0.36 | 50.05 | 110.74 | 7.55 | 15.38 |
| | Med. | 2.32 | 6.39 | 4.70 | 4.54 | 31.42 | 0.06 | 7.15 | 1.73 | 0.36 | 50.05 | 110.74 | 7.55 | 15.38 |
| | Std. | 0.33 | 1.54 | 2.42 | 3.19 | 7.18 | 0.00 | 1.94 | 0.20 | 0.36 | 18.59 | 42.99 | 0.27 | 0.93 |

### 3.1.1. pH Value

The pH value in the groundwater is mainly governed by $CO_2$ and $CO_3$. The pH values in Precambrian metamorphic rock range from 6.59 to 8.13, with an average value of 7.34; in Mesozoic granite, they range from 6.43 to 8.19, with an average value of 7.36; in Ogcheon metamorphic rock, they range from 6.73 to 8.10, with an average value of 7.56; and in Jeju volcanic rock, they range from 7.35 to 7.74, with an average value of 7.55, indicating a weak acidic to weak alkaline condition. The pH measurements for the majority of mineral waters in each geology were close to neutral, with an average of 7.34 to 7.56.

### 3.1.2. Electrical Conductivity (EC)

The EC value indicates the amount of material dissolved in groundwater [20]. There are large variations in EC values, not only in the samples collected from the different geologies, but also in the samples collected from the same geology. The EC range of Precambrian metamorphic rock is 161.98 to 349.53 µS/cm, with an average of 161.37 µS/cm; in Mesozoic granite, it is 86.80 to 221.42 µS/cm, with an average of 140.31 µS/cm; in Ogcheon metamorphic rock, it is 119.18 to 353.50 µS/cm, with an average of 264.12 µS/cm; and in Jeju volcanic rock, it is 80.34 to 141.14 µS/cm, with an average of 110.74 µS/cm. Jeju volcanic rock has the lowest EC because Jeju island mainly consists of permeable volcanic rocks, which do not allow an effective water-rock interaction due to the rapid penetration and water flow.

### 3.1.3. Temperature

Mineral water temperature according to each geology ranges from 8.03 to 18.92 °C, with an average of 14.33 to 15.38 °C, and a median of 14.44 to 15.38 °C. The temperature order of each geology is as follows: Jeju volcanic rocks (15.38 °C) > Mesozoic granite (15.11 °C) > Ogcheon metamorphic rocks (14.54 °C) > Precambrian metamorphic rocks (14.33 °C).

### 3.1.4. Cation Concentration

The ions in mineral water are primarily dissolved in the groundwater from certain chemical species in minerals that behave independently or together [21]. Consequently, the characteristics of the chemical species can be utilized to infer the behavior of ions in the groundwater. The mean value of total cations is highest in the mineral water of Ogcheon metamorphic rocks, and cation concentrations are in order of abundance, Ca > Na > Mg > K.

The $SiO_2$ concentrations in Precambrian metamorphic rock range from 6.97 to 32.50 mg/L, with an average of 18.09 mg/L, Mesozoic granite is 8.65 to 41.76 mg/L, with an average of 21.84 mg/L, Ogcheon metamorphic rock is 9.97 to 31.28 mg/L, with an average of 15.43 mg/L, and Jeju volcanic rock is 26.34 to 36.50 mg/L, with an average of 31.42 mg/L. The mean values, in order of abundance, are as follows: Jeju volcanic rocks > Mesozoic granite > Precambrian metamorphic rocks > Ogcheon metamorphic rocks. The behavior of silica in solutions is strongly influenced by the solubility of quartz, amorphous silica, and Si-bearing minerals [22–24].

The concentration range of Ca in Precambrian metamorphic rock is 4.33 to 48.84 mg/L, with an average of 18.69 mg/L; in Mesozoic granite, it is 7.67 to 34.43 mg/L, with an average of 15.70 mg/L; in Ogcheon metamorphic rock, it is 21.82 to 50.58 mg/L, with an average of 34.61 mg/L; and in Jeju volcanic rock, it is 2.99 to 6.40 mg/L, with an average of 4.70 mg/L. The mean values, in order of abundance, are as follows: Ogcheon metamorphic rocks > Precambrian metamorphic rocks > Jurassic granite > Jeju volcanic rocks. Ca is commonly enriched in calcite, dolomite, gypsum, apatite, and Ca-bearing silicate minerals.

The Na concentrations in Precambrian metamorphic rocks range from 2.07 to 17.8 mg/L, with an average of 7.60 mg/L; in Mesozoic granite, from 2.86 to 18.70 mg/L, with an average of 8.02 mg/L; in Ogcheon metamorphic rock, from 3.63 to 13.60 mg/L, with an average of 6.77 mg/L; and in Jeju volcanic rock, from 5.30 to 7.49 mg/L, with an average of 6.39 mg/L. Na is easily leached from albite plagioclase, which is one of the most common components in granites.

The range of K concentrations in Precambrian metamorphic rock is 0.49 to 2.75 mg/L, with an average of 1.29 mg/L; in Mesozoic granite, it is 0.10 to 1.67 mg/L, with an average of 0.64 mg/L; in Ogcheon metamorphic rock, it is 0.65 to 1.88 mg/L, with an average of 1.20 mg/L; and in Jeju volcanic rock, it is 2.09 to 2.56 mg/L, with an average of 2.32 mg/L. The mean values, in order of abundance, are as follows: Jeju volcanic rocks > Precambrian metamorphic rock > Ogcheon metamorphic rocks > Jurassic granite.

The Mg concentration range in Precambrian metamorphic rock is 0.41 to 12.23 mg/L, with an average of 3.36 mg/L; in Mesozoic granite, it is 0.33 to 5.65 mg/L, with an average of 1.84 mg/L; in Ogcheon metamorphic rock, it is 2.47 to 18.20 mg/L, with an average of 9.23 mg/L; and in Jeju volcanic rock, it is 2.28 to 6.79 mg/L, with an average of 4.54 mg/L. The mean values, in order of abundance, are as follows: Ogcheon metamorphic rocks > Jeju volcanic rocks > Precambrian metamorphic rock > Jurassic granite. The mean Mg value is the highest in Ogcheon metamorphic rocks because they are rich in minerals such as dolomite, high Mg-calcite, biotite, amphibole, and olivine.

### 3.1.5. Anion Concentrations

Total anion concentrations are the highest in groundwater of Ogcheon metamorphic rocks. Among anions, $HCO_3$ is the most abundant species, followed by $SO_4$ > Cl > F > $NO_3$.

The $HCO_3$ value range in Precambrian metamorphic rock is 17.08 to 197.25 mg/L, with an average of 79.39 mg/L; in Mesozoic granite, it is 34.53 to 127.34 mg/L, with an average of 62.60 mg/L; in Ogcheon metamorphic rock, it is 74.42 to 170.80 mg/L, with an average of 127.55 mg/L; and in Jeju volcanic rock, it is 36.91 to 63.20 mg/L, with an average of 50.05 mg/L. $HCO_3$ is the most abundant species in groundwater with neutral pH conditions [25].

The Cl concentration range in Precambrian metamorphic rock is 1.51 to 14.57 mg/L, with an average of 4.43 mg/L; in Mesozoic granite, it is 1.33 to 37.30 mg/L, with an average of 5.53 mg/L;

in Ogcheon metamorphic rock, it is 2.42 to 19.23 mg/L, with an average of 6.77 mg/L; and in Jeju volcanic rock, it is 5.78 to 8.52 mg/L, with an average of 7.15 mg/L.

The $SO_4$ concentrations in Precambrian metamorphic rock range from 1.63 to 30.67 mg/L, with an average of 8.72 mg/L; in Mesozoic granite, from 1.0 to 40.40 mg/L, with an average of 9.32 mg/L; in Ogcheon metamorphic rock, from 3.14 to 80.32 mg/L, with an average of 25.59 mg/L; and in Jeju volcanic rock, from 1.59 to 1.87 mg/L, with an average of 1.73 mg/L. Cl and $SO_4$ are considered important inorganic constituents of groundwater, which may deteriorate the quality of drinking water at higher extents [26].

The F concentrations in Precambrian metamorphic rock range from 0.05 to 1.12 mg/L, with an average of 0.35 mg/L; in Mesozoic granite, from 0.07 to 1.65 mg/L, with an average of 0.52 mg/L; in Ogcheon metamorphic rock, from 0.05 to 0.73 mg/L, with an average of 0.25 mg/L; and in Jeju volcanic rock, the F concentration is 0.06 mg/L.

The nitrate nitrogen ($NO_3$-N) concentration range in Precambrian metamorphic rock is 0.01 to 4.06 mg/L, with an average of 0.90 mg/L; in Mesozoic granite, it is 0.04 to 4.33 mg/L, with an average of 1.02 mg/L; in Ogcheon metamorphic rock, it is non-detectable to 2.24 mg/L, with an average of 0.87 mg/L; and in Jeju volcanic rock, it is 0.11 to 0.62 mg/L, with an average of 0.36 mg/L. The drinking water quality standard for nitrate nitrogen in South Korea is 10 mg/L [26].

According to the results of the mineral water quality analysis, seven manufacturers corresponding to 11.2% of the total number, have $NO_3$-N concentrations of 2 mg/L or more (Table 1). $NO_3$-N originates from soil with a significant amount of organic matter or polluted surface water, rather than being dissolved in groundwater via a water–rock reaction, and can be used as an indicator of contamination [27,28]. Considering that the mineral water companies in Korea are located in a mountainous area without pollution, the concentration values are high. It is likely that sub-surface water has been contaminated by inflow into wells due to insufficient casing and grouting. The drinking water quality standard for $NO_3$-N in Korea is 10 mg/L [3]. Although these seven manufacturers are well within the standard, the establishment of contamination management plans regarding $NO_3$-N, including environmental surveys around the wells and the installation of a pollution prevention facility using a shielding grouting system, are needed to improve the mineral water quality for people's health.

### 3.2. Permission and Production Amounts of Mineral Water

In South Korea, an environmental impact assessment, which includes a geological survey, geophysical exploration, and a water quantity/quality assessment, needs to be completed to apply for permission for a mineral water exploitation. If the environmental impact assessment is favorable, a permit effective for five years is issued. After the five years has elapsed, mineral water manufacturers can apply for an extension for another five years [3]. The DWMA limits a permitted extraction volume on a "per day" basis.

As of the end of December 2019, 62 mineral water manufacturers were registered in South Korea: 16 manufacturers were located in Gyeonggi Province, followed by 11 in Gyeongnam Province, 8 in Gangwon Province, 7 in Chungnam Province, 4 each in Chungbuk, Gyeongbuk, and Jeonnam Provinces, 3 in Jeonbuk Province, 2 each in Jeju and Ulsan Provinces, and 1 in Sejong Province. Most manufacturers in Gyeonggi Province are concentrated in the suburbs of the Seoul metropolitan area to meet the high demands of mineral water in that region. Many manufacturers are also distributed in Chungnam Province and Chungbuk Province because logistics costs are relatively low due to the advantage of a nationwide travel distance.

The total permitted amount of extracted mineral water in South Korea is 42,645 m³/day, of which Gyeonggi province accounts for the largest amount, at 12,837 m³/day, 30% of the total permitted amount (Table 3). The next is Gyeongnam Province with 8366 m³/day and the least is Sejong Province with 620 m³/day. Eleven companies with a daily permitted volume of 1000 m³/day or more account for 18% of the total number of manufacturers. Twenty-two companies with 500~1000 m³/day account for 35% of the total factories, while the remaining 47% produce less than 500 m³/day of mineral water.

Jeju Province Development Co. has the largest permitted volume of 3700 m$^3$/day, followed by Whain company with 3130 m$^3$/day, and Backhak company with 2564 m$^3$/day. Chilgabsan and Geumcheon, with 60 m$^3$/day and 70 m$^3$/day, respectively, have the lowest daily permitted extraction amount.

**Table 3.** The status of mineral water manufactures in the provinces in South Korea.

| Province | Numbers of Manufacturers | Daily Permitted Volume (m$^3$/day) | Daily Production Amounts (m$^3$/day) | Production Rate (%) |
|---|---|---|---|---|
| Chungbuk | 4 | 2752 | 1565 | 57 |
| Chungnam | 7 | 3969 | 982 | 25 |
| Gangwon | 8 | 3087 | 771 | 25 |
| Gyeonggi | 16 | 12,837 | 4638 | 36 |
| Gyeongbuk | 4 | 2710 | 458 | 17 |
| Gyeongnam | 11 | 8366 | 1736 | 21 |
| Jeonbuk | 3 | 2160 | 500 | 23 |
| Jeonnam | 4 | 1490 | 157 | 11 |
| Jeju | 2 | 3800 | 2948 | 78 |
| Sejong | 1 | 620 | 82 | 13 |
| Ulsan | 2 | 854 | 335 | 39 |
| Total | 62 | 42,645 | 14,172 | 33 |

The total daily actual production in South Korea is 14,172 m$^3$/day, which only represents 33% of the total permitted amount (Table 3). Gyeonggi Province covers the largest production of 4638 m$^3$/day, which is 25% of the total permitted volume. Next, Jeju Province has a production of 2948 m$^3$/day, representing 78%, and Sejong Province has the lowest production rate of 82 m$^3$/day, representing 13%. Jeju Province is the local government with a high production rate of 78% compared with the permitted volume. Although Jeju Province has only two factories, namely Jeju Samdasu Co. and Korea Airports Co., Jeju produces a large amount, close to 80% of the permitted volume, which is significantly higher than other factories.

Pumped source water is variously used as mineral water, washing water, and cleaning water. Around 70–75% of the water intake in most factories is used to produce mineral water, which is a suitable amount compared to the pumped water amount. Approximately 25–30% of the pumped source water is used for washing and the remaining 2% is used for other purposes, such as cleaning.

The actual production of all provinces except Jeju Province is less than 50% of the permitted amount, indicating that the optimal yield estimated in the environmental impact assessment may have been overestimated as more the mineral water manufacturers' actual optimal yield. This result suggests that it this a serious problem in quantity, which can cause a depletion of mineral water resources and deterioration of water quality.

*3.3. Market of Mineral Water*

The sales volume of the mineral water market has increased remarkably over the past 25 years, from 471,514 m$^3$/day in 1995, to 1,429,970 m$^3$/day in 2000, 3,362,100 m$^3$/day in 2010, and 4,770,722 m$^3$/day in 2018 (Table 4). The size of the market for mineral water increased 10-fold over the same period: from USD 60.5 million in 1995, to USD 130.2 million in 2000, USD 301.7 million in 2010, and USD 574.9 million in 2018 (Table 4). Jeju Samdasu, the most popular brand among domestic products, accounts for 40% of the domestic market [29].

**Table 4.** The status of the mineral water market in South Korea.

| Year | Sales Volume (m³/day) | Sales Amount (Million Dollar) | Sales Volume Increase (%) | Sales Amount Increase (%) |
|---|---|---|---|---|
| 1995 | 471,514 | 60.5 | | |
| 1996 | 893,002 | 114.9 | 189.4 | 189.9 |
| 1997 | 873,678 | 87.8 | 185.3 | 145.1 |
| 1998 | 940,356 | 75.3 | 199.4 | 124.5 |
| 1999 | 1,147,982 | 106.2 | 243.5 | 175.5 |
| 2000 | 1,429,970 | 130.2 | 303.3 | 215.2 |
| 2001 | 1,851,234 | 169.1 | 392.6 | 279.5 |
| 2002 | 2,024,537 | 179.6 | 429.4 | 296.9 |
| 2003 | 1,970,204 | 156.7 | 417.8 | 259.0 |
| 2004 | 2,259,584 | 174.3 | 479.2 | 288.1 |
| 2005 | 2,441,586 | 186.6 | 517.8 | 308.4 |
| 2006 | 2,550,832 | 214.4 | 541.0 | 354.4 |
| 2007 | 2,726,058 | 231.7 | 578.1 | 383.0 |
| 2008 | 3,049,812 | 256.3 | 646.8 | 423.6 |
| 2009 | 3,251,537 | 272.4 | 689.6 | 450.2 |
| 2010 | 3,362,100 | 301.7 | 713.0 | 498.7 |
| 2011 | 3,094,669 | 357.9 | 656.3 | 591.6 |
| 2012 | 3,253,001 | 388.1 | 689.9 | 641.5 |
| 2013 | 3,543,000 | 433.3 | 751.4 | 716.2 |
| 2014 | 3,358,498 | 478.9 | 712.3 | 791.6 |
| 2015 | 3,611,683 | 535.2 | 766.0 | 884.6 |
| 2016 | 4,359,408 | 564.5 | 924.6 | 933.1 |
| 2017 | 4,711,975 | 571.9 | 999.3 | 945.3 |
| 2018 | 4,770,722 | 574.9 | 1011.8 | 950.2 |

Mineral water was officially allowed to go on sale in 1995 and, in 1996, sales approximately doubled compared with the previous year. However, the growth of the sales amount slowed or declined until 1998 due to the foreign exchange crisis in 1997, before increasing again from 1999. The sales volume was 2.44 million m³/day in 2005 and 3.36 million m³/day in 2010, showing 10- and 7-fold increases, respectively, from 0.47 million m³/day in 1995, when mineral water sales were first permitted. On the other hand, the sales amounts were USD 186.6 million in 2005 and USD 301.7 million in 2010, which represent 3- and 5-fold increases, respectively, from USD 60.5 million in 1995. The relationship between the sales amount and sales volume, from when the domestic sales of mineral water were permitted in South Korea until the present, indicates that the mineral water market only grew quantitatively until 2013 due to its low-cost sales method, which is a problem in the market of mineral water, caused by the explosion of manufacturer numbers and excessive competition amongst them.

Twenty manufacturers, which account for 32% of the total number of 62, produce more than 100 m³/day, and 70% of the mineral water market is accounted for by eight large manufacturers. The average selling price of a 0.5 L bottle of domestic mineral water is USD 0.17, which is about one-fifth of the price of USD 0.83 for imported products. This phenomenon indicates that domestic mineral water is weak in competitiveness in comparison with foreign mineral water.

The Korea Herald estimated that the total sales of the mineral water market will amount to USD 890 in 2020 [30]. However, considering the sales amounts of USD 575 million in 2018, this seems difficult to achieve.

*3.4. Integrated Management System of the Mineral Water*

KIGAM analyzes and manages the data transmitted in real-time from mineral water manufacturers, and notifies and guides manufacturers if anomalous data are detected (Figure 4).

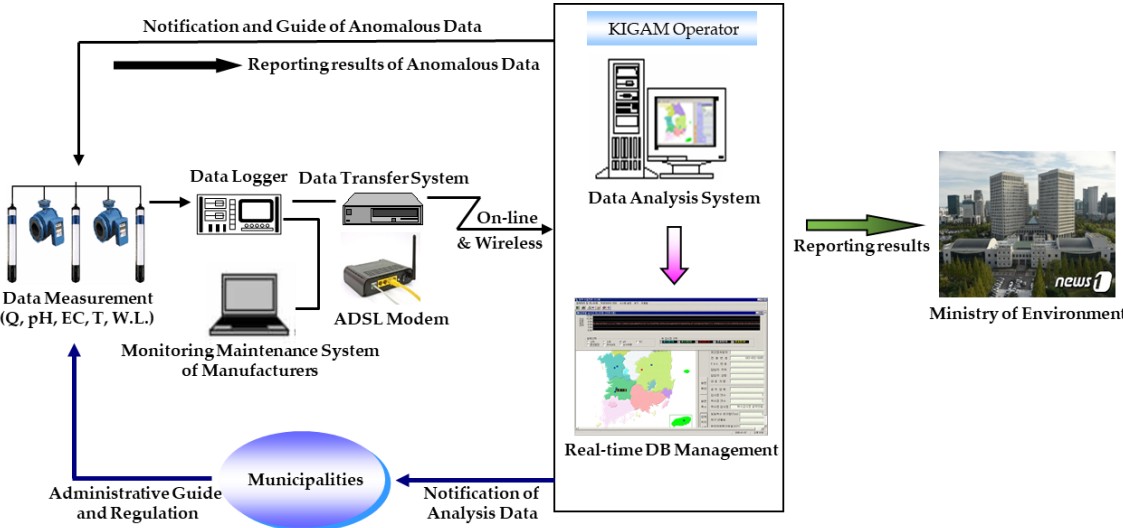

**Figure 4.** Management system of mineral water in South Korea.

### 3.4.1. IMS Building

The location of the manufacturer is plotted on the initial screen of the IMS based on GIS (Figure 2). The basic environment of the IMS is divided into the control button domain, map display domain, manufacturer's brief information domain, map control domain, metadata domain, and attribute data domain (Figure 5). The control button domain is used to control the manufacturer information of the vector type by the GIS, which is composed of map enlargement and reduction, domain movement, full area view, manufacturer's information view, and system shutdown button.

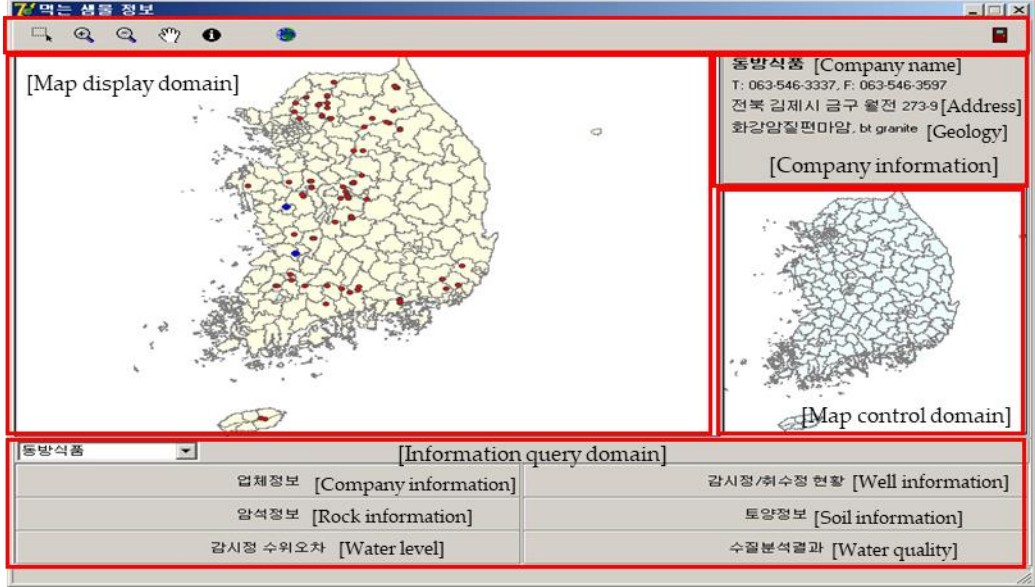

**Figure 5.** The basic environment of the integrated management system (IMS) for the mineral water management.

The map display domain shows the administrative district of the vector type (including city, county, and district), and the locations of the manufacturers. The map domain is controllable in the control button domain and the map control domain.

The manufacturer's brief information domain shows the manufacturer name, address, telephone/fax number, and geology, and is automatically updated when a manufacturer is selected on the map display domain or on the metadata and attribute data domain.

The map control domain is designed separately in order to easily move the viewpoint to a specific position when the map is zoomed in or out. In the map control domain, the portion displayed in the map is designated by a rectangle with red color. The size of the rectangle is changed according to the enlargement/reduction ratio of the map area, and the designated area can be moved by dragging.

The metadata and attribute domains are used to display the corresponding information of the manufacturers. In this domain, a specific manufacturer can be selected in the map area and the manufacturer's information can be selected separately from the map display domain. In addition, detailed information can be browsed through the panel of manufacturer information, the status of observation and pumping wells, rock/soil information, the error of groundwater level on the observation wells, and water quality analysis data.

### 3.4.2. Information Query

Inquiries about the present situation of manufacturers can be made in two ways. The first method is to click the info button on the control domain and select the company in the map domain; if it is difficult to select because the company is adjacent, the company can be selected using the control button and map control. Second, it is possible to select the company using the company name pull-down menu at the top of the attribute domain. The pull-down menu is designed to be automatically updated when the company is selected in the map domain.

The company status query is designed so that the contents for every kind of attribute information related to the corresponding company can be expressed in various forms by browsing the metadata information.

The status of monitoring and pumping wells includes information about the number of monitoring and pumping wells of the manufacturers. The inquiry method for the status of the monitoring and pumping wells is the same as the query method for the manufacturer information. This inquiry is designed so that the contents of attribute information related to the well can be updated from the metadata search when an inquiry of manufacturer information is carried out.

The groundwater level of the monitoring well is composed of the measurement date, automatically monitored data, and actual measurement data. Since the number of monitoring wells varies from company to company, the attribute information and chart area of the monitoring well information are controlled according to the number of monitoring wells. The attribute information is constructed as a separate database for each manufacturer and is displayed in the attribute information table as well as the chart through the company name inquiry. Automatically monitored water levels and actually measured water levels can be selected and displayed on the chart for each monitoring well.

The water quality database is made up of chemical components, such as K, Na, Ca, Mg, Fe, $SiO_2$, Al, As, and $NO_3$-N, as well as pH, EC, T, and the analysis date. Water quality data are stored as an individual database for each manufacturer and additional data can be inputted when further information is obtained. The results of the water quality analysis are provided in the form of a table in the order of analysis date and are shown on the chart for each chemical component through additional inquiry.

Rock and soil information includes the analysis results of rock and soil. The results of analysis are composed of separate databases, with each database linked to metadata. The constituents of the analysis are $SiO_2$, $Al_2O_3$, $Fe_2O_3$, MgO, MnO, $TiO_2$, $P_2O_5$, CaO, $K_2O$, $Na_2O$, etc. The results of analysis are displayed in the chart by company, and the components shown in the chart can be updated through additional inquiry.

### 3.4.3. Assessment of IMS

KIGAM, the groundwater specialized agency designated by Korea's Ministry of Environment, has operated an IMS to store and efficiently manage data transmitted from the mineral water manufacturers and help the manufacturers' technical difficulties. However, the role of IMS until the present has been limited to data storage and management, and efforts to find the problems of IMS and mineral water manufacturers by various analyses using these data have been insufficient. The problems of IMS are that a lot of information should be newly loaded in accordance with the current designed form, and the current IMS is not compatible with other operating systems and only runs on the Microsoft Windows operating system.

In order to preserve the quantity and quality of mineral water, it is considered that it needs to develop a groundwater flow model and a pollutant movement module that can be linked with the current IMS in the future, and to arrange a basis for predicting the change in groundwater around the manufacturers.

## 4. Conclusions

This study presented the current state of South Korean mineral water in terms of geology, production, market characteristics, water quality, and IMS, to determine a suitable strategy. First, we assessed the current status of mineral water in South Korea and identified a problem, then devised a strategy to solve the problem and improve the management of manufacturers.

Of particular note, there are no mineral water plants in sedimentary rock areas such as the Cretaceous Gyeongsang basin located in southeastern Korea, according to the geological setting data of mineral water manufacturers. The reason is believed to be that most of the geology of this area is composed of sedimentary rocks with high total dissolved solids. However, Evian is a global mineral water brand located in a sedimentary rock area with high total dissolved solids. Thus, we propose that a roadmap for the development of sedimentary rock areas should be produced to compete with global mineral water brands such as Evian.

The Korean mineral water market has mainly grown quantitatively due to low-cost sales methods since domestic sales began. Mineral water manufacturers should progress from quantitative growth by low-cost sales, and establish their own specialization plans according to geological characteristics, so that they can create high value-added products. By doing so, mineral water manufacturers can enhance their competitiveness in the world market and grow into international brands. Awareness among mineral water manufacturers regarding specialization plans should be increased.

Seven mineral water manufacturers have $NO_3$-N concentrations of 2 mg/L or more. Therefore, we propose the establishment of contamination management plans regarding $NO_3$-N in order to improve the mineral water quality.

The IMS was developed in order to efficiently analyze and manage various types of information transmitted in real-time from the manufacturers. The current IMS has a shortcoming in that a large amount of information should be newly inputted in accordance with the designed form. Therefore, it is necessary to design an independent editing system of the attribute information, and then link this information with the mineral water IMS. The current IMS also only runs on the Microsoft Windows operating system, so we suggest that support for other operating systems should be planned for the future.

This study is unique as a policy study, with aims to improve the national competitiveness of mineral water by the efficient management of mineral water in South Korea. In conclusion, we propose the mineral water exploitation of sedimentary rock areas, development of specialization plans according to geological characteristics, environmental surveys around wells, installation of a pollution prevention facility, design of an independent editing system for attribute information, and support for other operating systems in addition to Microsoft Windows. In addition to these, we propose that policy makers establish a management plan for mineral water every 5 to 10 years for the conservation of mineral water resources.



**Author Contributions:** Conceptualization, B.D.L. and S.-Y.H.; Data curation, B.D.L., Y.H.O., W.B.K., J.H., W.-R.L.; Formal analysis, B.D.L., S.-Y.H., Y.H.O., W.B.K., J.H., S.-J.C., and W.-R.L; Investigation, B.D.L., Y.H.O., W.B.K., J.H., and W.-R.L; Methodology, B.D.L. and S.-Y.H.; Project administration, S.-Y.H.; Resources, B.D.L.; Software, B.D.L.; Supervision, S.-Y.H.; Validation, B.D.L.; Visualization, B.D.L, Y.H.O., W.B.K., J.H., S.-J.C., and W.-R.L; Writing—original draft, B.D.L. and S.-Y.H.; Writing—review and editing, B.D.L. and S.-Y.H. All authors have read and agreed to the published version of the manuscript.

**Funding:** This research was financially supported by the Basic Research Project (203411), the Korea Institute of Geoscience and Mineral Resources (KIGAM) funded by the Ministry of Science and ICT (MSICT) of Korea as well as by the National Research Foundation of Korea (NRF) grant (No. NRF-2020R1A2B5B02002198) funded by the MSICT of Korea.

**Conflicts of Interest:** The authors declare no conflict of interest.

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
