# Peer review of "State and Strategy of Production, Market and Integrated Management of Mineral Water, South Korea"

_water, doi:10.3390/w12061615_

Round 1

Reviewer 1 Report

Dear Authors

I propose some minor changes:

Line 14: Please change and use trough out the text full name of your country -South Korea-. The term "Korea" could be misunderstood as Korean peninsula as a whole (with North Korea).

Line 76: "to present state of the"

Please add what kind of state? Economic, quality, quantity.

I would also propose that the aim sentence is broken down on two or three sentences. Now is rather long.

On of the sentences could start like this: The aim was achieved through/by...

Line 117: Change title to: "Geological map of South Korea and locations mineral water bottlers facilities"

In would not say that they manufacture the water as they just abstract and bottle the water.

I would also propose to make an overview map o Korean peninsula smaller and geological map larger. It would be beneficial if certain landmarks would be added - major 3-4 cities.

Line 119:

Please change the title 2.2 Methods into 2.2 Database preparation (here you present how the data was obtained)

Please add chapters:

2.3 Statistical methods

-explain statistical methods used to present data on chemical characteristics and market)

2.4 Integrated management system

  • explain the methods used in presenting IMS. I would say that chapter 3.4 could be as whole moved into methods under 2.4. Text is just explaining already established knowledge. Or not? Were you - authors or your institution the developers?
  • If so then separate between methods on which the IMS was established (GIS, SQL, KIGAM) (chapter 2.4) and on what was developed and could be presented as a result (functional IMS) (Figure 4, 5) (Chapter 3.4). The result is the management system, the process of recording and transmitting data. Is data publicly available?

2.5 Uncertainties and shortcomings

Please write some self-reflection. What could be done better if some other data, methods, processes would be used.

Line 139: Please add the discussion to the 3. Results and Discussion. Chapter 4 should be only Conclusions.

Line 139: On general the discussion is the weakest part of this manuscript. I would expect from authors to compare the results on chemical analysis and development of IMS to similar studies worldwide, regional or national.

Enhance the discussion. Authors could form additional subchapter 3.5 Assessment of IMS, and discuss practical usefulness, usefulness to science, policy, user, companies, public. What could be upgraded?

I am not totally sure if authors managed to present problem by using data on chemical characteristics. These data must support the aim and offer some conclusions on why IMS is so important. For example subchapter 3.1.5 Anion Concentration or others. Authors do not give any conclusions or discuss the results. Is it bad, is it good? What do this result show in supporting IMS?

Line 438-441: Conclusions

Authors did asses the current quality and quantity status. Although I am not able to identify the problem. Authors must discuss this matter in the text. What is the problem? Currently they present in the results chapter only facts, pure data without discussion.

I am not sure if you identified the problem. You need to discuss the problem in the text. What is the problem?

I am not sure if the authors presented in the text how they devised a strategy to solve the problem. This needs to be more clearly stated in the text.

For sure they presented how they improved the management of mineral water bottlers. I would say that South Korea has now an exemplary system of real-time remote sensing of mineral water quality and quantity.

Line 456: Problem is identified here at the end of the manuscript. This should be discussed already when results are presented, or even in the case study description. Think about that.

Line 476: Please link this conclusion with the results. On the base of which results did authors propose exploitation of sedimentary rock areas, etc.

Please ansvere in the conclusions chapter to these questions:

Why is this research unique?

What are the advantages and Shortcomings/uncertainties of this research?

What did we (the scientific community) learn from it?

What are the benefits for stakeholders (companies, public)?

What are your recommendations for policy-makers?

Future work?

Reviewer 2 Report

The manuscript "State and Strategy of Production, Market and Integrated Management of Mineral Water, South Korea" present the results of extensive investigation of the origin, quality and extraction of mineral waters in South Korea.

The manuscript also present the principle for mineral water monitoring in South Korea by Integrated Management System (IMS).

The manuscript is well written, and obtained results are clearly presented.

I find this manuscript interesting as a potential role model of systematic mineral water management.

Therefore, I recommend the acceptance of this manuscript after minor corrections. Suggestions for corrections are give in attached pdf file.
